# Using digital health tools for the Remote Assessment of Treatment Prognosis in Depression (RAPID): a study protocol for a feasibility study

Valeria de Angel [ID],[1,2] Serena Lewis,[1,3] Sara Munir,[4] Faith Matcham,[1] Richard Dobson,[2,5] Matthew Hotopf[1,2]

For numbered affiliations see end of article.

**Correspondence to**
Valeria de Angel;
valeria.de_angel@kcl.ac.uk

## ABSTRACT

**Introduction** Digital health tools such as smartphones and wearable devices could improve psychological treatment outcomes in depression through more accurate and comprehensive measures of patient behaviour. However, in this emerging field, most studies are small and based on student populations outside of a clinical setting. The current study aims to determine the feasibility and acceptability of using smartphones and wearable devices to collect behavioural and clinical data in people undergoing therapy for depressive disorders and establish the extent to which they can be potentially useful biomarkers of depression and recovery after treatment.

**Methods and analysis** This is an observational, prospective cohort study of 65 people attending psychological therapy for depression in multiple London-based sites. It will collect continuous passive data from smartphone sensors and a Fitbit fitness tracker, and deliver questionnaires, speech tasks and cognitive assessments through smartphone-based apps. Objective data on sleep, physical activity, location, Bluetooth contact, smartphone use and heart rate will be gathered for 7 months, and compared with clinical and contextual data. A mixed methods design, including a qualitative interview of patient experiences, will be used to evaluate key feasibility indicators, digital phenotypes of depression and therapy prognosis. Patient and public involvement was sought for participant-facing documents and the study design of the current research proposal.

**Ethics and dissemination** Ethical approval has been obtained from the London Westminster Research Ethics Committee, and the Health Research Authority, Integrated Research Application System (project ID: 270918). Privacy and confidentiality will be guaranteed and the procedures for handling, processing, storage and destruction of the data will comply with the General Data Protection Regulation. Findings from this study will form part of a doctoral thesis, will be presented at national and international meetings or academic conferences and will generate manuscripts to be submitted to peer-reviewed journals.

**Trial registration number** https://doi.org/10.17605/OSF.IO/PMYTA

## STRENGTHS AND LIMITATIONS OF THIS STUDY

⇒ The current mixed methods design to evaluate feasibility and acceptability will provide a deeper understanding of the associations between patterns of missing data across the different data collection methods and clinical state.

⇒ Both passive sensing and active validated questionnaire-based data collection methods will be evaluated.

⇒ A 7-month participant follow-up provides a picture of engagement in the longer term, compared with previous studies.

⇒ For pragmatic reasons, the study uses a non-randomised, non-controlled design, which will limit conclusions about digital changes to treatment response.

⇒ This study does not use devices that are validated for medical use, drawing instead from digital sensors in Android smartphones and a Fitbit fitness tracker, which have been previously used in mental health research.

## INTRODUCTION

Depression is a leading cause of disability worldwide,[1] yet response to treatment is poor, with only 50%–60% of people recovering after 3 months of treatment.[2 3] Mental health science relies almost exclusively on subjective self-report to diagnose mental illness and measure outcomes. Reliance is therefore placed on patients being able to accurately recall and communicate complex mood states during clinical interviews, which many patients find difficult.[4 5] Depending on subjective methods introduces a vulnerability to recall biases[6] which may worsen with increased severity.[7]

The use of digital tools within mental health has the potential to enhance traditional self-report measures by improving aspects of symptom tracking, illness management and treatment support. Remote measurement

technologies (RMTs) such as smartphones and wearable devices can unobtrusively capture a more accurate picture of a patient's clinical state in a continuous way, with far less burden to the user. Through embedded sensors, they can detect changes in behaviours associated with depressive symptomatology such as sleep,[8] sociability,[9] physical activity[10] and speech.[11] Detecting such changes in a person's behaviour can provide invaluable information for tailoring and improving treatment.

Given the relative recency of the field, and with an eye towards clinical implementation, studies on RMTs and depression have largely comprised proof-of-concept, feasibility and acceptability studies. While studies show RMTs to be generally feasible and acceptable,[12] the data predominantly come from small, non-clinical or student samples with a median follow-up time of 2 weeks.[13]

To our knowledge, no such feasibility studies have been published on the use of RMTs to track mood and behaviour in clinical populations undergoing psychotherapy for depression. This population could derive greater benefits from the application of such digital methods by alleviating distress in more potentially severe cases and reducing pressure on healthcare services. In addition to the barriers of adopting RMTs for mood monitoring found across populations, such as concerns around privacy, confidentiality, affordability and accessibility, there are likely to be additional considerations related to the help-seeking populations that remain unexplored and may help or hinder implementation practices in the future.

From a clinical perspective, exploring feasibility and acceptability in therapy populations can shed light on how changes in severity affect engagement with and use of digital devices, how they can be used to complement treatment and the barriers and facilitators to their use and implementation within services. From a methodological perspective, it would be important to establish the extent to which the amount and quality of data is usable and unbiased given that the added workload from therapy exercises, homework and clinical questionnaires as well as the severity of symptoms such as decreased motivation and cognitive abilities are likely to affect engagement and device use.[12] Such studies could also inform future projects about the likely uptake in these samples in order to establish sample sizes and allocate resources.

## Study aims

The primary aim of this project is to evaluate the extent to which data collection with digital tools on clinical samples is feasible. Specifically, the feasibility of using RMTs, such as Android smartphones and a Fitbit wearable device, to collect behavioural and clinical data in people undergoing therapy for depressive disorders, to establish the extent to which they can be potentially useful biomarkers of depression and changes in clinical state. The purpose of this is to describe patterns of user engagement and missing data, which are likely to impact the scientific integrity of future large-scale studies.

Secondary aims are to identify candidate signals for digital biomarkers by detecting correlations between objective features and clinical characteristics, and to explore whether these signals have prognostic value in the context of psychological treatments.

## METHODS AND ANALYSIS

### Design

This is an observational, prospective cohort study of people attending psychological therapy for depression. It will use RMTs to gather active and passive data for up to 7 months and will adopt a mixed methods design to evaluate the feasibility and acceptability of such data collection methods.

### Setting

Participants will be drawn from Improving Access to Psychological Therapies (IAPT) services in South London. The participating IAPT services will be from the boroughs of Lambeth, Lewisham and Croydon within the South London and Maudsley NHS Foundation Trust. IAPT is a publicly funded self-referral outpatient programme providing evidence-based psychological treatments for adults with mild-to-moderate mental health disorders. The service is free at the point of delivery.

Recruitment for this study initiated in June 2020 during the COVID-19 pandemic. Given the government-imposed travel restrictions and social distancing measures, the entirety of this study will be carried out remotely.

### Sample size

Formal sample size calculations are not a requirement for feasibility studies[14]; however, the general recommendation is for samples of 50–60 participants to assess feasibility outcomes.[15] In order to address our secondary aims, a sample size of 50 would be sufficiently powered to detect a correlation coefficient of 0.39 and above, assuming a significance level of 0.05 and type 2 error value of 0.20. Based on previous studies, we expect such an effect size.[16–18] To account for a potential attrition rate of 20% we will aim to recruit 65 participants.

### Recruitment

IAPT clinicians will act as gatekeepers to the initial recruitment process. Patients within their service, who have previously agreed to be contacted for research purposes, will be invited to take part, either by phone call or email. They will be given a summary of the aims and procedures of the study and be screened for eligibility as per the inclusion/exclusion criteria below. When screening is done by email, participants receive a personalised email with a description of the study and a link to an online screening tool that participants can complete in their own time, the responses of which are relayed to the research team. If willing and eligible, potential participants will be sent the participant

information sheet and given at least 24 hours before going through the consent procedures and being enrolled in the study.

## Inclusion criteria

A. Adults with a current depressive episode as measured by the Mini International Neuropsychiatric Interview (MINI[19]).

B. Being on the waiting list to receive treatment for depression at IAPT services, with an expected wait of at least 7 days (to a maximum of 5 weeks) between scheduled enrolment and first treatment session. Due to the prevalent comorbidity with anxiety disorders, people with a main diagnosis of anxiety were also included, provided they met inclusion criterion A.

C. Existing ownership of Android smartphone with sufficient memory space for the relevant apps.

D. Able and willing to use a wrist-worn device for duration of the study.

E. Able to give informed consent for participation.

F. Sufficient English language skills to understand consent process and questionnaires.

## Exclusion criteria

A. Lifetime diagnosis of bipolar disorder, schizophrenia and schizoaffective disorders as these have different digital patterns to depression.[20 21]

B. Health anxieties that may significantly worsen with constant monitoring of behaviour.

C. Extensive sharing of smartphone with friends or family.

D. Night shifts, pregnancy or living with a baby aged 0–6 months (due to sleep disruptions).

## Study procedures

Once interest and eligibility have been ascertained, participants will be invited to attend an enrolment session via video call. Figure 1 shows the study timeline for participants as they enter the study. After a further review of study procedures and opportunity for questions, participants will be asked to sign an electronic consent form. Consent can be taken either using the Qualtrics platform, which has been approved for this purpose, or via MS Word or PDF documents which participants electronically sign from their devices.

### Enrolment/baseline

Following consent, the enrolment session comprised three further sections: (1) obtaining sociodemographic and clinical data, (2) completion of self-reported questionnaires, and (3) technology set-up. The researchers will take demographic and clinical information related to current and previous physical and mental health conditions, family history, treatment status as well as phone use, previous experience with health apps and devices and social and physical activity levels. In order to detect the presence of a depressive episode, and define whether atypical in nature, MINI and the Atypical Depression Diagnostic Scale[22] will be administered. At the end of the session, participants will be asked to complete a battery of self-reported questionnaires as shown in table 1.

### Technology set-up

Participants will be asked to download four apps on their phone: RADAR passive RMT (pRMT) app, which collects background sensor data from smartphones; the RADAR active RMT (aRMT) app, which delivers clinical questionnaires; THINC-it for Remote Assessment of Disease and Relapse-Central Nervous System (RADAR-CNS), an app assessing cognitive function; and the Fitbit app. These will be linked in call to the RADAR base platform.[23] A Fitbit Charge 3 or 4 is then delivered to them within one to two working days, at which point they are guided through the set-up. Participants will be given £10.00 for completing the enrolment session and keep the Fitbit after the study. Table 1 shows the schedule of events for the Remote Assessment of Treatment Prognosis in Depression study.

### Follow-up

From enrolment, longitudinal collection of active and passive data begins. The current study will use the RADAR base platform and their apps to collect passive and active data, as well as a Fitbit API integration source.[23] More information can be found at radar-base.org/.

### Passive measures

Passive measures will be continuously gathered from smartphone sensors via the pRMT app and wearable sensors from the Fitbit. Sensor data will include Global Positioning System (GPS), acceleration, light, phone interaction (total time on phone and app usage), paired and nearby Bluetooth devices, number of saved contacts,

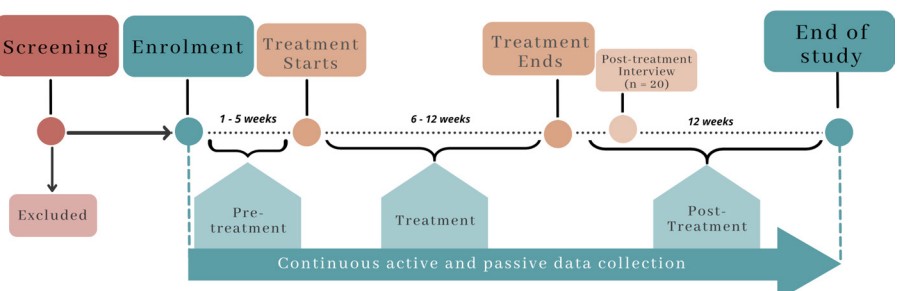

**Figure 1** Study timeline for participants.

**Table 1**  Schedule of events

| Event | Enrolment | Questionnaire frequency | | | |
|---|---|---|---|---|---|
| | | Weekly | Fortnightly | Monthly | Endpoint |
| Informed consent | x | | | | |
| Sociodemographic data | x | | | | |
| Clinical history | x | | | | |
| MINI | x | | | | |
| ADDS | x | | | | |
| Smartphone apps set-up | x | | | | |
| **Active measures\* (from REDCap)** | | | | | |
| *Validated questionnaires* | | | | | |
| SAPAS | x | | | | |
| BIPQ | x | | | | x |
| Life stress (SRRS) | x | | | | x |
| CTQ | x | | | | |
| AUDIT | x | | | | |
| PHQ–9 | x | | x | | x |
| GAD-7 | x | | x | | x |
| Rumination (RRS) | x | | | x | |
| AUDIT (short version) | x | | x | | |
| Oslo 3-item Social Support Scale | x | | | x | |
| Perceived Stress Scale | x | | | x | x |
| WAI-SR† | | | | | x |
| *Contextual information* | | | | | |
| Caffeine intake | x | x | | | |
| Treatment status and content‡ | x | x | | | |
| Social activities | x | | x | | |
| Social distancing practices | x | | x | | |
| COVID-19 experience | x | | | x | |
| **Active measures\* (from aRMT app)** | | | | | |
| QIDS-SR | x | x | | | |
| WSAS | x | x | | | x |
| Speech task | x | | x | | |
| Perceived sleep§ | x | x | | | |
| Cognition (THINC-it app) | x | | | x | |
| **Passive measures** | | | | | |
| Fitbit Charge | To be worn throughout the study. | | | | |
| pRMT app: GPS, acceleration, light, phone interaction, Bluetooth devices, number of contacts, battery level, weather | Will run in the background gathering data from enrolment. | | | | |
| **Qualitative interview†** | | | | | x |

\*Weekly time spent completing questionnaires should not exceed 10 min.
†Completed once during treatment.
‡Only for the duration of treatment.
§Daily for 90 days.
ADDS, Atypical Depression Diagnostic Scale; aRMT, active RMT; AUDIT, Alcohol Use Disorder Identification Test; BIPQ, Brief Illness Perceptions Questionnaire; CTQ, Childhood Trauma Questionnaire; GAD-7, Generalized Anxiety Disorder; GSP, Global Positioning System; MINI, Mini International Neuropsychiatric Interview; PHQ-9, Patient Health Questionnaire; pRMT, passive RMT; QIDS-SR, Quick Inventory of Depressive Symptomatology-Self-Report; REDCap, Research Electronic Data Capture; RRS, Rumination Response Scale; SAPAS, Standardised Assessment of Personality: Abbreviated Scale; SRRS, Social Readjustment Rating Scale; WAI-SR, Working Alliance Inventory-Short Revised; WSAS, Work and Social Adjustment Scale.

battery level and weather. Fitbit generates digital features relating to sleep, physical activity and heart rate.

Neither the Fitbit nor the apps are validated medical tools, as they are not intended to diagnose or treat a medical condition; RADAR-based apps are purpose built for research, while the Fitbit is marketed as a fitness tracker. Despite questions surrounding the ability of digital sensors in detecting the behaviours of interest accurately, they have been found to reliably detect sleep, physical activity and location.[24–26]

No personally identifiable information will be gathered from these sensors; GPS signals are obfuscated and relative to previous location rather than exact points, and no contact details, website or app content is collected by the apps. Personal privacy is thus protected, and no identification of an individual's home address or precise geographical location can be gathered.

## Active measures

Participants will be asked to respond to questionnaires throughout the study period. Some of them will be delivered via the aRMT app, others will be collected via the Research Electronic Data Capture (REDCap) software,[27] a web-based platform for research that sends email notifications to participants throughout the study.

A. Weekly emailed questionnaires: participants will receive weekly emails with a link to complete REDCap-delivered questionnaires, which can be complete on a smartphone or a computer. Questionnaires will be scheduled at different time intervals (fortnightly, monthly), in such a way that the maximum amount of time needed to complete them is 10 min/week.

B. Weekly aRMT tasks: the aRMT app is designed to collect health information from research participants by sending them notifications and asking them to complete in-app tasks and questionnaires. These will include questions on depression, functionality, subjective sleep experiences and a speech task.

C. Speech task: participants will be asked to undertake two speech tasks. The first task will require them to read out prewritten text, and the second task will ask them to answer out loud a question such as: 'Can you describe something you are looking forward to this week?'. Participants will record their voice in quiet surroundings for both tasks via the aRMT app. Acoustic features such as pitch, jitter, shimmer, formants and intensity will be extracted.

D. Cognition via THINC-it app: once a month, the aRMT app will notify participants that it's time to complete the THINC-it tasks, they will be asked to open the THINC-it app to do so. THINC-it is a validated tool designed to assess cognitive function in depression.[28] The tests incorporated in this tool—the One-Back Test, the Trail Making Test Part B, the Digit Symbol Substitution Test, Choice Reaction Time Task—assess attention, processing speed, executive function, learning and memory. The tool also incorporates the Per-

ceived Deficits Questionnaire,[29] a self-report questionnaire that assesses a person's cognitive concerns.

## End of study

Twelve weeks after participants have finished treatment, the research team will contact them to finalise their time in the study and complete endpoint assessments. In case of an unexpected change regarding their treatment, such as treatment being reduced or extended, the 3-month follow-up will commence on the day of their last core treatment session with IAPT services.

## Extra participant contact

To maintain engagement and stay abreast of any issues, participants will be contacted after their first week after enrolment, and then in months 1, 3, 5 and 7. Researchers will initially contact participants on the phone unless an alternative method of contact is preferred and followed up with an email. Any issues raised in these calls, or sporadically reported by participants throughout the course of the study, will be recorded. Additionally, participants will be sent a monthly newsletter, via email, which will include study updates, tech tips and any frequently asked questions.

## Post-treatment qualitative interview

In order to inform the feasibility and acceptability aims of this study, participants who complete therapy will be invited to take part in an optional qualitative interview. It will be a 30 min semistructured interview looking at participant experiences of using RMTs during psychotherapy for depression. We will invite participants to this interview once they complete treatment and will interview the first 20 who agree to take part. See online supplemental materials for a full interview schedule.

## Outcome measures
### Primary outcomes

The primary outcome is to establish the feasibility of using wearable devices and smartphone sensors to monitor the behaviour of people with depression while receiving psychological treatment. Key feasibility outcomes will be related to clinical and methodological considerations, and will evaluate recruitment and participant flow, subjective reports of acceptability of methods, data availability and data quality.

The following feasibility outcomes will be reported:

► Estimates of recruitment and attrition rates (figure 2).
► Presence and absence of passive data: 'wear time' for wearable devices and 'on time' for smartphone sensors, and the extent to which the available data allow for correlational and predictive analyses with significant statistical power.
► Active data availability and data quality: percentage number of tasks completed.
► Qualitative data: participant experience and attitudes towards data collection instruments and procedures.

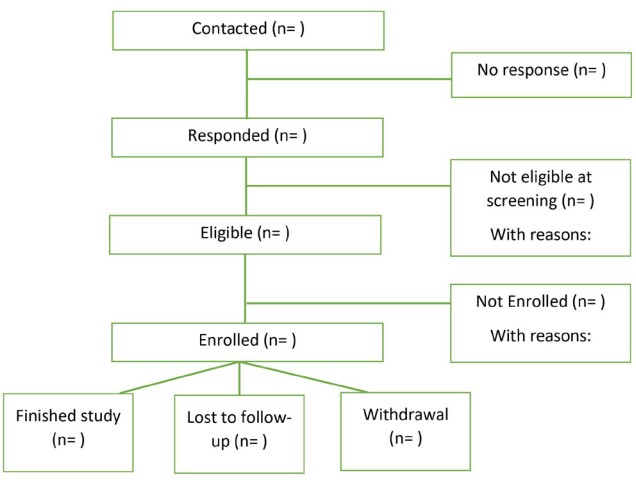

**Figure 2** Participant flow in the Remote Assessment of Treatment Prognosis in Depression (RAPID) study.

## Secondary outcomes

Secondary outcomes will evaluate the relationship between digital data and clinical outcome measures, both at individual time points and as prognostic factors for recovery after treatment. Scores on clinical scales will be used in addition to individual symptom domains.

Digital outcomes will be derived from smartphone and Fitbit sensors, and digital features from passive measures described in the 'Passive Measures' section, such as GPS signal patterns, sleep, phone use and Bluetooth interactions. They will be extracted to form averages that encapsulate daily, weekly and within-treatment means and SDs or frequency counts.

Clinical state will be measured using:
► Patient Health Questionnaire (PHQ-9)[30]: the nine-item questionnaire that is widely used for measuring depression in IAPT services.
► Generalized Anxiety Disorder[31] questionnaire: will be measured as it gathers important anxiety symptoms which are so often comorbid with depression.
► Quick Inventory of Depressive Symptomatology-Self-Report (QIDS-SR)[32]: this is a 16-item inventory of depression for patients who identify as depressed or who may be suffering from depression.
► Work and Social Adjustment Scale[33]: a measure of quality of life/disability, which is a five-item assessment of perceived social and work-related functional impairment used widely across a range of mental and physical disorders.
► Cognition via the THINC-it app.

Participants will be considered to be in remission if they have experienced a reduction of at least 50% in depressive symptomatology from the start of treatment, or no longer meet criteria for depression according to the PHQ-9 (scoring below the cut-off of 5). Subdimensions of depression—for example, interest in activities, motivation, appetite—will be gathered from the QIDS-SR.

The sociodemographic, clinical and contextual variables measured, such as illness severity, cognitive function and social support, will be taken as covariates.
► Standardised Assessment of Personality: Abbreviated Scale[34]: an eight-item personality test that screens for personality disorder.
► Brief Illness Perceptions Questionnaire[35]: provides an insight into the participant's views about their underlying condition and how well they see themselves coping with it.
► Life Stress Scale (Social Readjustment Rating Scale)[36]: this is a retrospective questionnaire for identifying major stressful life events.
► Childhood Trauma Questionnaire[37]: a 26-item scale that assesses five types of maltreatment: sexual abuse, physical abuse, emotional neglect, physical neglect and emotional abuse.
► Alcohol Use Disorder Identification Test[38]: widely used scale in primary care that measures alcohol consumption, drinking behaviours, and identifies harmful alcohol use.
► Oslo 3-item Social Support Scale[39]: a brief instrument that assesses social support.
► Perceived Stress Scale[40]: a measure of the degree to which situations in one's life are appraised as stressful. Items were designed to assess how unpredictable, uncontrollable and overloaded respondents find their lives to be.
► Working Alliance Inventory-Short Revised[41]: a 12-item scale measuring therapeutic alliance on three key components: (a) agreement on the tasks of therapy, (b) agreement on the goals of therapy, and (c) development of an affective bond.
► Rumination Response Scale[42]: a 10-item scale that measures rumination.
► COVID-19-related questions: isolation status and their perceived effect, confirmed or suspected diagnosis of COVID-19 as well as social distancing practices.
► Treatment status: questions on whether psychotherapy has begun, whether they take concomitant medication and adherence (eg, 'Are you taking medication for your mental health? If so, which one?' or 'Have you missed any psychotherapy sessions this week?') and the broad content of their therapy sessions.

## STATISTICAL ANALYSIS
### Primary aims

Since our primary aims are descriptive, they will be presented as frequencies, percentages, means and SDs as appropriate. Missing data will be calculated as the percentage amount of data available from the total amount of expected data. Engagement will be assessed through data availability in passive data streams and data quality will be measured as the number of active tasks that are incomplete. Associations between engagement and clinical characteristics will be explored. Recordings of the semistructured interviews will be transcribed verbatim, checked for accuracy by a second researcher and analysed

using a deductive approach to thematic analysis, with the iterative categorisation technique.[43] Where participant responses can be quantified, they will be presented as aggregated counts or percentages, otherwise, summaries of participant responses will be presented narratively. Sporadic reports of issues with the technology or study methodologies will be summarised.

### Secondary aims

Digital features that account for sleep, activity, sociability and cognition will be extracted from sensor data and correlated against scores on scales of depression, anxiety and functionality. Feature extraction will replicate the methods used by the RADAR-CNS consortium.[23 44] Regression or classification analyses will be carried out for clinical scores to see whether higher impairment is associated with behavioural features. Regression and classification approaches will also be used to determine whether clinical data predict subject attrition or missing data patterns.

We will carry out univariate and multivariate associations on digital features and clinical state, as well as within and between individual comparisons. In order to unearth digital profiles in the sample, individuals will be clustered together based on their response patterns using latent class analysis. This person-centred approach will unpick some of the heterogeneity in the sample and assumes there are underlying latent variables that underpin distinct symptom profiles,[45] and has been used extensively in the construction of the subtypes of depression.[46] This model will aid in the description of longitudinal behavioural patterns in this sample.

To evaluate the prognostic value of digital features, we will use machine learning methods on the extracted aggregated features and clinical information, provided there are sufficient data points. A multivariate prediction model will be constructed, and different feature selection algorithms will be applied. Model performance will be evaluated through cross-validation, putting stress on sensitivity and specificity of relapse prediction model. We will also use dynamic structural equation modelling to evaluate the lagged associations across study time points.

Where data are missing at random and assuming it is not significantly high, multiple imputation methods will be carried out. If missing data are high, this may be incorporated into the model as a predictor, or otherwise used informatively.

### Reporting standards

In the interest of open and reproducible science, we will follow basic transparency recommendations,[13 47–49] including the reporting of basic demographic and clinical data, attrition and participation rates, missing data, evidence of the validity or reliability of the sensors and devices used. For each behavioural feature, a full definition and description of feature construction will be provided, with links to GitHub repositories and source code, where available. Definition and handling of missing data will be specified. In machine learning models, model selection strategy, performance metrics and parameter estimates in the model with CIs, or non-parametric equivalents, will be described in full.

### Patient and public involvement

This research was reviewed by a team with experience of mental health problems and their carers who have been specially trained to advise on research proposals and documentation through the Feasibility and Acceptability Support Team for Researchers: a free, confidential service in England provided by the National Institute for Health Research Maudsley Biomedical Research Centre via King's College London and South London and Maudsley NHS Foundation Trust.

## ETHICS AND DISSEMINATION

This study has been reviewed and given favourable opinion by the London Westminster Research Ethics Committee, approval from the Health Research Authority, Integrated Research Application System (project ID: 270918) and confirmation of capacity and capability to carry out research from the South London and Maudsley NHS Foundation Trust. The research will be carried out in accordance with the Helsinki Declaration and International Conference on Harmonisation-Good Clinical Practice Guidelines. Privacy and confidentiality will be guaranteed and the procedures for handling, processing, storage and destruction of the data will comply with the General Data Protection Regulation. Data collected will be hosted on KCL infrastructure. Participant Fitbit accounts will be created using generic email accounts so no personal details are shared with Fitbit.

The results of the study will be presented at local, national and international meetings or academic conferences, and will generate manuscripts to be submitted to peer-reviewed journals. Additionally, the results from this study will form part of a doctoral thesis and will be shared with participants, if they wish, after the study has been completed.

## DISCUSSION

If digital technologies are to fulfil their potential to revolutionise the clinical management of mental health conditions, we need to establish the feasibility and acceptability of using RMTs such as wearables and smartphones to track mood and behaviour in those seeking and undergoing treatment for such conditions.

There are some anticipated challenges faced by this study. Continuous tracking of behaviours like physical activity and sleep may result in favourable changes to health behaviours and improved self-management. While the current study is non-interventional and does not aim to affect improvement rates, such behavioural changes may directly impact mood and treatment outcome.

The main concern, however, arises from the impact of the COVID-19 pandemic. Although the current public

health crisis will impact this study in several ways, we identify three main areas. First, part of the data collection will cover periods of time when there were government-imposed restrictions to movement and social proximity, meaning people's daily routine will have been greatly disrupted, and signals will bear additional noise. Second, the impact on individuals' mental health will be sizeable.[50] The profile of patients referred to psychological services may be different than before or after the pandemic, as we are faced with new mental health challenges.[51] Finally, the pandemic has resulted in the forced adoption of digital technology for all aspects of life, including healthcare, likely affecting attitudes towards technology and therefore engaging with and accepting RMTs.[52]

Given the recency of the field and the interest in implementing digital technologies within healthcare, assessing the acceptability and feasibility of such methods in this target population is of great importance in informing implementation efforts as well as planning future research studies involving such samples. Through the use of mixed methods, the current study aims to identify and address as many of these issues as possible.

**Author affiliations**
¹Department of Psychological Medicine, Institute of Psychiatry, Psychology and Neuroscience, King's College London, London, UK
²NIHR Maudsley Biomedical Research Centre, South London and Maudsley NHS Foundation Trust, London, UK
³Department of Psychology, University of Bath, Bath, UK
⁴Lewisham Talking Therapies, South London and Maudsley NHS Foundation Trust, London, UK
⁵Department of Biostatistics and Health Informatics, Institute of Psychiatry, Psychology and Neuroscience, King's College London, London, UK

**Contributors** VdA, MH, RD and FM conceived and developed this project. VA, FM, SM and SL contributed to the design and implementation work within health services, as well as the acquisition of data. All authors contributed to the revision and edition of the manuscript and have provided their final approval of the current version.

**Funding** This study represents independent research funded by the National Institute for Health Research (NIHR) Biomedical Research Centre at South London and Maudsley NHS Foundation Trust and King's College London. RD is supported by the following: (1) NIHR Biomedical Research Centre at South London and Maudsley NHS Foundation Trust and King's College London, UK; (2) Health Data Research UK, which is funded by the UK Medical Research Council, Engineering and Physical Sciences Research Council, Economic and Social Research Council, Department of Health and Social Care (England), Chief Scientist Office of the Scottish Government Health and Social Care Directorates, Health and Social Care Research and Development Division (Welsh Government), Public Health Agency (Northern Ireland), British Heart Foundation and Wellcome Trust; (3) The BigData@Heart consortium, funded by the Innovative Medicines Initiative-2 Joint Undertaking under grant agreement number 116074. This joint undertaking receives support from the European Union's Horizon 2020 research and innovation programme and EFPIA; it is chaired by DE Grobbee and SD Anker, partnering with 20 academic and industry partners and ESC; (4) the National Institute for Health Research (NIHR) University College London Hospitals Biomedical Research Centre; (5) the NIHR Biomedical Research Centre at South London and Maudsley NHS Foundation Trust and King's College London; (6) the UK Research and Innovation London Medical Imaging and Artificial Intelligence Centre for Value Based Healthcare; (7) the NIHR Applied Research Collaboration South London (NIHR ARC South London) at King's College Hospital NHS Foundation Trust.

**Competing interests** MH is the principal investigator of the RADAR-CNS programme, a precompetitive public–private partnership funded by the Innovative Medicines Initiative and European Federation of Pharmaceutical Industries and Associations. The programme receives support from Janssen, Biogen, MSD, UCB and Lundbeck.

**Patient and public involvement** Patients and/or the public were involved in the design, or conduct, or reporting, or dissemination plans of this research. Refer to the Methods section for further details.

**Patient consent for publication** Not required.

**Provenance and peer review** Not commissioned; externally peer reviewed.

**ORCID iD**
Valeria de Angel http://orcid.org/0000-0002-5109-3636

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
