## [Reviewer comments · BMJ Open]

ARTICLE DETAILS

TITLE (PROVISIONAL)	Using Digital Health Tools for the Remote Assessment of Treatment Prognosis In Depression (RAPID): A Study Protocol for a feasibility study
AUTHORS	de Angel, Valeria; Lewis, Serena; Munir, Sara; Matcham, Faith; Dobson, Richard; Hotopf, Matthew

VERSION 1 – REVIEW

REVIEWER	Bulgheroni, Maria Ab.Acus srl, Milan, IT, R&D
REVIEW RETURNED	27-Jan-2022

GENERAL COMMENTS	Very nice and clear protocol. Only some general comments that might help in data analysis: - the technology setup includes several apps to be installed and run. In addition there are weekly emailed questionnaires to be completed using REDCap. This setup implies a medium/high level of digital literacy to deal with the different interfaces, requests and so on and this might impact on attrition rate. I suggest to add the collection of information describing the level of acquaintance to technology usage by participants (if not already included in acquired sociodemographic data).- Among the passive measures it is included GPS. GPS data may be a sensitive data. How do you manage it?- Passive measures are not fully described in Table 1, please detail them.- Among the feasibility outcome is not fully clear what do you mean by "amount of data necessary to conduct correlation and predictive analyses, please clarify.- Please better detail which "digital data" you will use to assess the relationship with clinical outcome measurements. While clinical scales are very well detailed, computed digital features are not sufficiently described.
---

REVIEWER	Tang, Xiangdong Sichuan University West China Hospital
REVIEW RETURNED	18-Feb-2022

GENERAL COMMENTS	The authors will launch an observational, prospective cohort study of 65 patients who attending psychological therapy for depression and anxiety in multiple London-based sites. It will collect continuous passive data from smartphone sensors and a Fitbit fitness tracker, and deliver questionnaires, speech tasks and cognitive assessments through smartphone-based apps. The authors will conduct a 7-month follow-up and a Qualitative Interview at the endpoint to determine the feasibility of using RMTs
---

	, and to identify candidate signals for digital biomarkers by detecting correlations between objective features and clinical characteristics, and to explore whether these signals have prognostic value in the context of psychological treatments. It is characteristic that the acoustic features such as pitch, jitter, shimmer, formants and intensity will be extracted for further analysis, and cognitive function data will be collected remotely. The following are some of the parts that need to be modified.  1. In the INTRODUCTION part, the authors state "However, the field is in its infancy, with the literature comprised of mostly small-scale studies with student populations outside of a clinical setting.", which is derived from the authors' summary of two Systematic Reviews. However, there have been a number of articles in recent years on the use of smartphones and wearable devices for patients with depression, other psychiatric disorders and somatic disorders. The authors should consider revising the presentation accordingly or adding new literature. 2. The authors will use smartphone sensors, mobile apps (including the RADAR passive RMT (pRMT) app, the RADAR active RMT (aRMT) app, THINC-it® for RADAR-CNS and the Fitbit app) and Fitbit Charge 3 or 4 for data collection, it needs to be stated whether they are validated medical tools or have been confirmed by medical trials. These needs to be stated in the protocol and noted in the Limitations section. 3. In Limitations, the limitations of observational studies without a control group should be described. 4. The first paragraph of the INTRODUCTION part mainly introduces depression, but does not introduce anxiety disorder. In STUDY AIMS, patients with depression are also taken as the observation objects, but in TITLE and METHODS AND ANALYSIS, the objects will attend therapy for depression and anxiety, which needs to be explained. 5. The demographic data collection protocol and the semi-structured qualitative interview outline should be listed. 6. Figure 1 needs further improvement, it is not clear enough and does not explain the whole research process. 7. SPIRIT checklist needs to be provided.
--	--

VERSION 1 – AUTHOR RESPONSE

REVIEWER 1's Comments to Author: Dr. Maria Bulgheroni

REVIEWER COMMENT: Very nice and clear protocol. Only some general comments that might help in data analysis:

- the technology setup includes several apps to be installed and run. In addition there are weekly emailed questionnaires to be completed using REDCap. This setup implies a medium/high level of digital literacy to deal with the different interfaces, requests and so on and this might impact on attrition rate. I suggest to add the collection of information describing the level of acquaintance to technology usage by participants (if not already included in acquired sociodemographic data).

AUTHOR RESPONSE:

Dear Dr. Bulgheroni – thank you very much for your compliments and comments on the protocol, they are greatly appreciated!

We attempt to bring the pre-requisite of digital literacy to a minimum, but we agree that it is likely to impact attrition to some extent. At the initial enrolment session, participants are set up with the technology with the researcher and given guidance on how to use it. Even though the passive apps need no input from the user, the active apps require the participant to be able to access their emails, select the link, and complete the online questionnaires on their phone. We have follow-up calls with participants where we check participant comfort with manipulating the technology. These calls happen within the first week, and then the first month of participation, after which any further notifications or requests follow a pattern they will now be familiar with.

We do not have a question on their level of comfort with technology (something we will no doubt add to future research!), but we do ask at baseline about previous experience with health apps and wearable devices. I have added a note on this on the paragraph "Enrolment/Baseline", under "Study Procedures".

REVIEWER COMMENT: Among the passive measures it is included GPS. GPS data may be a sensitive data. How do you manage it?

AUTHOR RESPONSE: GPS signals are obfuscated and relative to previous location rather than exact points, so we can see how a person moves, but not where they are on a map exactly.

- Passive measures are not fully described in Table 1, please detail them.

AUTHOR RESPONSE: Full passive data details have been added to table 1. We have also deleted step count as a passive measure derived from the app (modification in paragraph "Passive Measures"), as it will be derived from the Fitbit.

- Among the feasibility outcome is not fully clear what do you mean by "amount of data necessary to conduct correlation and predictive analyses, please clarify.

AUTHOR RESPONSE: Thank you for pointing this out. Since we expect there will be significant missing data, we would like to identify whether the amount of missing data will hinder statistical analysis with significant statistical power. This point has been clarified as follows:

"Presence and absence of passive data: 'wear time' for wearable devices and 'on time' for smartphone sensors, and the extent to which the available data allows for correlational and predictive analyses with significant statistical power."

- Please better detail which "digital data" you will use to assess the relationship with clinical outcome measurements. While clinical scales are very well detailed, computed digital features are not sufficiently described.

AUTHOR RESPONSE: An additional paragraph has been included under "Secondary Outcomes" with more detail on digital data. It includes a link to the "Passive Measures" section to avoid repetition. Further detail on feature extraction is mentioned "Statistical Analysis" > "Secondary Aims", where the reader is directed towards references 23 and 41.

REVIEWER 2's Comments to Author: Dr. Xiangdong Tang

REVIEWER COMMENT:

The authors will launch an observational, prospective cohort study of 65 patients who attending psychological therapy for depression and anxiety in multiple London-based sites. It will collect continuous passive data from smartphone sensors and a Fitbit fitness tracker, and deliver questionnaires, speech tasks and cognitive assessments through smartphone-based apps. The authors will conduct a 7-month follow-up and a Qualitative Interview at the endpoint to determine the feasibility of using RMTs , and to identify candidate signals for digital biomarkers by detecting correlations between objective features and clinical characteristics, and to explore whether these signals have prognostic value in the context of psychological treatments. It is characteristic that the acoustic features such as pitch, jitter, shimmer, formants and intensity will be extracted for further analysis, and cognitive function data will collected remotely.

The following are some of the parts that need to be modified.

1. In the INTRODUCTION part, the authors state "However, the field is in its infancy, with the literature comprised of mostly small-scale studies with student populations outside of a clinical setting.", which is derived from the authors' summary of two Systematic Reviews. However, there have been a number of articles in recent years on the use of smartphones and wearable devices for patients with depression, other psychiatric disorders and somatic disorders. The authors should consider revising the presentation accordingly or adding new literature.

AUTHOR RESPONSE:

Dear Dr. Tang – many thanks for reviewing this paper and providing such valuable feedback. I have attempted to address your comments below.

On your first point, we agree that there has been a growing body of research looking at digital health tools in clinical populations in recent years. We have therefore revised the presentation to the following:

“However, in this emerging field, most studies are small and based on student populations outside of a clinical setting.”

REVIEWER COMMENT: 2. The authors will use smartphone sensors, mobile apps (including the RADAR passive RMT (pRMT) app, the RADAR active RMT (aRMT) app, THINC-it® for RADAR-CNS and the Fitbit app) and fitbit Charge 3 or 4 for data collection, it needs to be stated whether they are validated medical tools or have been confirmed by medical trials. These needs to be stated in the protocol and noted in the Limitations section.

AUTHOR RESPONSE: This is a good point to include in our manuscript. We have therefore added the following paragraph under the “Passive Measures” section:

“Neither the Fitbit nor the apps are validated medical tools, as they are not intended to diagnose or treat a medical condition; RADAR-based apps are purpose-built for research, while the Fitbit is marketed as a fitness tracker. Despite questions surrounding the ability of digital sensors in detecting the behaviours of interest accurately, they have been found to reliably detect sleep, physical activity and location ²⁴⁻²⁶.”

Additionally, the following has been added to the “Strengths and Limitations” section:

- *“This study does not use devices that are validated for medical use, drawing instead from digital sensors in Android smartphones and a Fitbit fitness tracker, which have been previously used in mental health research.”*

REVIEWER COMMENT: 3. In Limitations, the limitations of observational studies without a control group should be described.

AUTHOR RESPONSE: The following has been added to the “Strengths and Limitations” section:

- *“For pragmatic reasons, the study uses a non-randomised, non-controlled design, which will limit conclusions about digital changes to treatment response.”*

REVIEWER COMMENT: 4. The first paragraph of the INTRODUCTION part mainly introduces depression, but does not introduce anxiety disorder. In STUDY AIMs, patients with depression are also taken as the observation objects, but in TITLE and METHODS AND ANALYSIS, the objects will attend therapy for depression and anxiety, which needs to be explained.

AUTHOR RESPONSE: The main focus of the study is on depression, and the main exclusion criteria involves experiencing a depressive disorder. Due to the prevalence of comorbid anxiety, we did not wish to exclude those with a primary diagnosis of anxiety (provided they still met criteria for depression) and decided to represent this in the description of the sample. We appreciate this could be a source of confusion, so we have amended the title and abstract. We have also clarified the inclusion criteria in the Recruitment section as follows:

“Inclusion criteria:

- a) Adults with a current depressive episode as measured by the Mini International Neuropsychiatric Interview (MINI).*
- b) Being on the waiting list to receive treatment for depression at IAPT services, with an expected wait of at least 7 days (to a maximum of 5 weeks) between scheduled enrolment and first treatment session. Due to the prevalent comorbidity with anxiety disorders, people with a main diagnosis of anxiety were also included, provided they met inclusion criteria a). “*

REVIEWER COMMENT: 5. The demographic data collection protocol and the semi-structured qualitative interview outline should be listed.

AUTHOR RESPONSE: The full treatment schedule for the qualitative interview will be included as a supplementary material. This has been signposted in the text under “Post-treatment Qualitative Interview”. Basic sociodemographic data is being collected; this is stated under section “Enrolment/Baseline”, and is listed in Table 1.

REVIEWER COMMENT: 6. Figure 1 needs further improvement, it is not clear enough and does not explain the whole research process.

AUTHOR RESPONSE: We have attempted to address this comment by modifying figure 1. This is a summary of study procedures, aimed at illustrating the order of study events and study timeline for participants. Should the reviewer require further changes, we would be happy to hear specific aspects that could be improved upon.

REVIEWER COMMENT: 7. SPIRIT checklist needs to be provided.

AUTHOR RESPONSE: We had not originally included a SPIRIT checklist because many of the items will not apply to non-interventional, observational studies. I would therefore not wish to have a SPIRIT checklist published alongside the current manuscript. In an attempt to address the concern that all

relevant aspects of a protocol are included, I have enclosed a SPIRIT checklist below with relevant page numbers for where each item is addressed on the manuscript.

SPIRIT 2013 Checklist: Recommended items to address in a clinical trial protocol and related documents*

Section/item	Item No	Description	Addressed on page
Administrative information			
Title	1	Descriptive title identifying the study design, population, interventions, and, if applicable, trial acronym	1
Trial registration	2a	Trial identifier and registry name. If not yet registered, name of intended registry	2
	2b	All items from the World Health Organization Trial Registration Data Set	NA
Protocol version	3	Date and version identifier	Document title
Funding	4	Sources and types of financial, material, and other support	9
Roles and responsibilities	5a	Names, affiliations, and roles of protocol contributors	1
	5b	Name and contact information for the trial sponsor	NA
	5c	Role of study sponsor and funders, if any, in study design; collection, management, analysis, and interpretation of data; writing of the report; and the decision to submit the report for publication, including whether they will have ultimate authority over any of these activities	9
	5d	Composition, roles, and responsibilities of the coordinating centre, steering committee, endpoint adjudication committee, data management team, and other individuals or groups overseeing the trial, if applicable (see Item 21a for data monitoring committee)	NA
Introduction			
Background and rationale	6a	Description of research question and justification for undertaking the trial, including summary of relevant studies (published and unpublished) examining benefits and harms for each intervention	3

	6b	Explanation for choice of comparators	NA
Objectives	7	Specific objectives or hypotheses	9
Trial design	8	Description of trial design including type of trial (eg, parallel group, crossover, factorial, single group), allocation ratio, and framework (eg, superiority, equivalence, noninferiority, exploratory)	3
Methods: Participants, interventions, and outcomes			
Study setting	9	Description of study settings (eg, community clinic, academic hospital) and list of countries where data will be collected. Reference to where list of study sites can be obtained	4
Eligibility criteria	10	Inclusion and exclusion criteria for participants. If applicable, eligibility criteria for study centres and individuals who will perform the interventions (eg, surgeons, psychotherapists)	4
Interventions	11a	Interventions for each group with sufficient detail to allow replication, including how and when they will be administered	NA
	11b	Criteria for discontinuing or modifying allocated interventions for a given trial participant (eg, drug dose change in response to harms, participant request, or improving/worsening disease)	NA
	11c	Strategies to improve adherence to intervention protocols, and any procedures for monitoring adherence (eg, drug tablet return, laboratory tests)	NA
	11d	Relevant concomitant care and interventions that are permitted or prohibited during the trial	NA
Outcomes	12	Primary, secondary, and other outcomes, including the specific measurement variable (eg, systolic blood pressure), analysis metric (eg, change from baseline, final value, time to event), method of aggregation (eg, median, proportion), and time point for each outcome. Explanation of the clinical relevance of chosen efficacy and harm outcomes is strongly recommended	7-8
Participant timeline	13	Time schedule of enrolment, interventions (including any run-ins and washouts), assessments, and visits for participants. A schematic diagram is highly recommended (see Figure)	4-6
Sample size	14	Estimated number of participants needed to achieve study objectives and how it was determined, including clinical and statistical assumptions supporting any sample size calculations	4

Recruitment	15	Strategies for achieving adequate participant enrolment to reach target sample size	4
-------------	----	---	---

Methods: Assignment of interventions (for controlled trials) NA

Allocation:

Sequence generation	16a	Method of generating the allocation sequence (eg, computer-generated random numbers), and list of any factors for stratification. To reduce predictability of a random sequence, details of any planned restriction (eg, blocking) should be provided in a separate document that is unavailable to those who enrol participants or assign interventions	
Allocation concealment mechanism	16b	Mechanism of implementing the allocation sequence (eg, central telephone; sequentially numbered, opaque, sealed envelopes), describing any steps to conceal the sequence until interventions are assigned	
Implementation	16c	Who will generate the allocation sequence, who will enrol participants, and who will assign participants to interventions	
Blinding (masking)	17a	Who will be blinded after assignment to interventions (eg, trial participants, care providers, outcome assessors, data analysts), and how	
	17b	If blinded, circumstances under which unblinding is permissible, and procedure for revealing a participant's allocated intervention during the trial	

Methods: Data collection, management, and analysis

Data collection methods	18a	Plans for assessment and collection of outcome, baseline, and other trial data, including any related processes to promote data quality (eg, duplicate measurements, training of assessors) and a description of study instruments (eg, questionnaires, laboratory tests) along with their reliability and validity, if known. Reference to where data collection forms can be found, if not in the protocol	5
	18b	Plans to promote participant retention and complete follow-up, including list of any outcome data to be collected for participants who discontinue or deviate from intervention protocols	NA

Data management	19	Plans for data entry, coding, security, and storage, including any related processes to promote data quality (eg, double data entry; range checks for data values). Reference to where details of data management procedures can be found, if not in the protocol	8-9
Statistical methods	20a	Statistical methods for analysing primary and secondary outcomes. Reference to where other details of the statistical analysis plan can be found, if not in the protocol	8
	20b	Methods for any additional analyses (eg, subgroup and adjusted analyses)	8
	20c	Definition of analysis population relating to protocol non-adherence (eg, as randomised analysis), and any statistical methods to handle missing data (eg, multiple imputation)	8

Methods: Monitoring

Data monitoring	21a	Composition of data monitoring committee (DMC); summary of its role and reporting structure; statement of whether it is independent from the sponsor and competing interests; and reference to where further details about its charter can be found, if not in the protocol. Alternatively, an explanation of why a DMC is not needed	NA
	21b	Description of any interim analyses and stopping guidelines, including who will have access to these interim results and make the final decision to terminate the trial	9
Harms	22	Plans for collecting, assessing, reporting, and managing solicited and spontaneously reported adverse events and other unintended effects of trial interventions or trial conduct	NA
Auditing	23	Frequency and procedures for auditing trial conduct, if any, and whether the process will be independent from investigators and the sponsor	NA

Ethics and dissemination

Research ethics approval	24	Plans for seeking research ethics committee/institutional review board (REC/IRB) approval	9
Protocol amendments	25	Plans for communicating important protocol modifications (eg, changes to eligibility criteria, outcomes, analyses) to relevant parties (eg, investigators, REC/IRBs, trial participants, trial registries, journals, regulators)	

Consent or assent	26a	Who will obtain informed consent or assent from potential trial participants or authorised surrogates, and how (see Item 32)	4
	26b	Additional consent provisions for collection and use of participant data and biological specimens in ancillary studies, if applicable	NA
Confidentiality	27	How personal information about potential and enrolled participants will be collected, shared, and maintained in order to protect confidentiality before, during, and after the trial	6
Declaration of interests	28	Financial and other competing interests for principal investigators for the overall trial and each study site	10
Access to data	29	Statement of who will have access to the final trial dataset, and disclosure of contractual agreements that limit such access for investigators	9
Ancillary and post-trial care	30	Provisions, if any, for ancillary and post-trial care, and for compensation to those who suffer harm from trial participation	NA
Dissemination policy	31a	Plans for investigators and sponsor to communicate trial results to participants, healthcare professionals, the public, and other relevant groups (eg, via publication, reporting in results databases, or other data sharing arrangements), including any publication restrictions	9
	31b	Authorship eligibility guidelines and any intended use of professional writers	
	31c	Plans, if any, for granting public access to the full protocol, participant-level dataset, and statistical code	8-9
Appendices			
Informed consent materials	32	Model consent form and other related documentation given to participants and authorised surrogates	
Biological specimens	33	Plans for collection, laboratory evaluation, and storage of biological specimens for genetic or molecular analysis in the current trial and for future use in ancillary studies, if applicable	NA

VERSION 2 – REVIEW

REVIEWER	Bulgheroni, Maria Ab.Acus srl, Milan, IT, R&D
-----------------	--

REVIEW RETURNED	18-Mar-2022
GENERAL COMMENTS	I am fine with the reviewed manuscript. This kind of studies is strongly needed to push the adoption of digital monitoring technologies in medicine and I would like to thank the authors for their contribution.
REVIEWER	Tang, Xiangdong Sichuan University West China Hospital
REVIEW RETURNED	31-Mar-2022
GENERAL COMMENTS	The authors have solved the issues.